# A Comparative Phylogeography of Three Marine Species with Different PLD Modes Reveals Two Genetic Breaks across the Southern Caribbean Sea

**DOI:** 10.3390/ani13152528

**Published:** 2023-08-05

**Authors:** Juan Carlos Narváez-Barandica, Julián F. Quintero-Galvis, Juan Carlos Aguirre-Pabón, Lyda R. Castro, Ricardo Betancur, Arturo Acero Pizarro

**Affiliations:** 1Centro de Genética y Biología Molecular, Universidad del Magdalena, Carrera 32 No 22–08, Santa Marta 470004, Colombia; jaguirrep@unimagdalena.edu.co (J.C.A.-P.); lcastro@unimagdalena.edu.co (L.R.C.); 2Instituto de Ciencias Ambientales y Evolutivas, Facultad de Ciencias, Universidad Austral de Chile, Valdivia 5110566, Chile; julianquintero1924@gmail.com; 3Biology Department, University of Oklahoma, Norman, OK 73019, USA; ricardo.betancur@ou.edu; 4Instituto de Estudios en Ciencias del Mar (CECIMAR), Universidad Nacional de Colombia sede Caribe, Santa Marta 470006, Colombia; aacerop@unal.edu.co

**Keywords:** biogeographic barrier, phylogeographic break, conservation genomics, Caribbean Sea

## Abstract

**Simple Summary:**

The comparative phylogeography of marine species with contrasting dispersal potential across the southern Caribbean Sea was evaluated by the presence of two putative barriers: the Magdalena River plume (MRP) and the combination of the absence of a rocky bottom and the almost permanent upwelling in the La Guajira Peninsula (ARB + PUG). Samples of each species were collected in five locations from Capurganá to La Guajira. For the first time, sufficient evidence of a phylogeographic break caused by the MRP is provided, mainly for *Acanthemblemaria rivasi*, a fish in a coral reef. The ARB + PUG barrier causes another break for *A. rivasi* and *Cittarium pica* (rocky shore mollusk species). We identified three populations for *A. rivasi* and *C. pica* from five locations, while *Nerita tessellata* presented one population. *Acanthemblemaria rivasi* and *C. pica* fit the hierarchical population model and share a similar phylogeographic history. Our results show how the biological traits of these three species and the biogeographic barriers have influenced their phylogeographic structure. Finally, we discussed why the Santa Marta and La Guajira marine sectors are essential for conserving marine species across the southern Caribbean Sea.

**Abstract:**

The comparative phylogeography of marine species with contrasting dispersal potential across the southern Caribbean Sea was evaluated by the presence of two putative barriers: the Magdalena River plume (MRP) and the combination of the absence of a rocky bottom and the almost permanent upwelling in the La Guajira Peninsula (ARB + PUG). Three species with varying biological and ecological characteristics (i.e., dispersal potentials) that inhabit shallow rocky bottoms were selected: *Cittarium pica* (PLD < 6 days), *Acanthemblemaria rivasi* (PLD < 22 days), and *Nerita tessellata* (PLD > 60 days). We generated a set of SNPs for the three species using the ddRad-seq technique. Samples of each species were collected in five locations from Capurganá to La Guajira. For the first time, evidence of a phylogeographic break caused by the MRP is provided, mainly for *A. rivasi* (AMOVA: Φ_CT_ = 0.420). The ARB + PUG barrier causes another break for *A. rivasi* (Φ_CT_ = 0.406) and *C. pica* (Φ_CT_ = 0.224). Three populations (*K* = 3) were identified for *A. rivasi* and *C. pica*, while *N. tessellata* presented one population (*K* = 1). The Mantel correlogram indicated that *A. rivasi* and *C. pica* fit the hierarchical population model, and only the *A. rivasi* and *C. pica* comparisons showed phylogeographic congruence. Our results demonstrate how the biological traits of these three species and the biogeographic barriers have influenced their phylogeographic structure.

## 1. Introduction

The ocean is considered a continuous environment and an open system. This implies that marine species exhibit genetic connectivity throughout their distribution due to the currents influencing gene flow between populations [1]. However, most studies have shown that marine species exhibit a level of population genetic subdivision in response to historical, geological, or ecological–oceanographic factors, although in different ways in taxa with contrasting life histories [1,2,3,4,5,6,7,8,9,10,11]. Combining these factors with biological information (e.g., reproductive biology, functional traits, and morphological adaptations) and ecological information allows us to understand marine organisms’ phylogeographic and genetic structure [2,5,6,9,12,13,14,15,16]. There is an ongoing discourse within the scientific community regarding the interrelationship between marine species’ genetic structure and their aptitude to disperse [2,6,17,18,19,20], which is predominantly influenced by their pelagic larval duration (PLD). This correlation is multifaceted and thus, the impact of larval duration on gene flow remains a contentious subject. Furthermore, other variables such as habitat availability, resource availability, and species interactions can also significantly influence a species’ genetic structure [21,22]. For instance, disparate genetic populations may emerge due to differences in available habitats, and the promotion or impediment of gene flow between populations can be impacted by species interactions or those interactions of the individuals during different life stages with the environment [23,24,25]. These conditions are crucial during natural selection and can affect the genetic structure. Nevertheless, a plethora of studies have indicated that species with extensive pelagic larval lifespans tend to display reduced substructuring or panmixia [11,13,26,27,28,29,30], while species that exhibit a PLD of a few days or lack pelagic larvae [26,27] tend to present phylogeographic breaks or population genetic substructuring in the presence of contemporary or historical barriers [6,10,13]. These barriers involve hydrological processes such as marine currents and gradients in the physicochemical properties of seawater due to continental river discharges and upwelling zones. Other factors include the absence of specific habitats and variation in shoreline geomorphology, including large distances that limit the dispersal of adult or larval stage organisms [5,6,7,9,11,31,32].

The marine sector of Colombia is in the south region of the Caribbean Sea [33,34] and presents different characteristics on its coasts that allow for the evaluation of different biogeographical hypotheses [17,34,35,36]. This marine sector has been characterized by various historical events, such as changes in the geomorphology of the coastline due to variations in sea level during the last glaciation and the uplift and movement of Caribbean basin mountain systems, such as the Sierra Nevada de Santa Marta [35]. This movement changed the continental shelf, the direction of marine currents, and the location of the mouth of the Magdalena River [37], which may have affected the phylogeographic structure of the marine–coastal organisms. Indeed, various phylogeographic studies in the Caribbean region used different genetic markers for some species with different larval dispersal times and suggested the existence of a break in the genetic connectivity of populations on both sides of the Caribbean, principally among the Central Caribbean and West Indies locations [17,23,38,39,40,41,42,43]. This genetic break is placed between Venezuela and Colombia and may be caused by the Magdalena River plume (MRP). However, several studies have rejected this hypothesis because the sampling design was unsuitable to test for this phylogeographic or genetic break. For example, some studies have not considered locations in Colombia [39,40,42,43,44], and those that did have only sampled close to one side of the river [41,45]. In other cases, the studies sampled both sides but only considered species of pelagic larvae that were more than 12 days in duration and concluded that the Magdalena River plume is not a barrier and the Caribbean Current (CC) and the Caribbean Counter Current (CCC) were the major factors causing genetic connectivity across the Colombian Caribbean i.e., coral, shrimp, mollusk, sea urchin, and fish species [29,41,46,47,48,49,50,51,52,53,54]. Therefore, further investigations are necessary to evaluate the MRP effect on other marine species.

A second barrier is believed to be located at 74–71° W, including Santa Marta and the La Guajira Peninsula [17,35,36], which is attributed to the almost permanent upwelling [55]. The predominant oceanic currents in the area include the CC and the CCC. These currents play a crucial role in the upwelling process. The CC, which flows from east to west, transports warm surface waters toward the Gulf of Venezuela and central Caribbean Sea. Meanwhile, the CCC derives from the Panama–Colombia Gyre [55,56], and during the weakening of the north trade winds (June–November), the current influences the transport of a mass of water up to the area of Cabo de la Vela [55,57,58]. As these currents converge near the Guajira Peninsula, the combined effect of offshore winds and the Coriolis effect causes an upward movement of the deeper, nutrient-rich waters to the surface. The winds blowing parallel to the coastline push the surface waters offshore, allowing the colder, nutrient-rich waters to rise from the depths. The La Guajira Peninsula experiences a strong upwelling from December to May due to high-intensity trade winds. However, the effect is weaker from June to August. It is important to note that the weak upwelling is mainly concentrated at the northern end of the Peninsula, specifically at Cabo de la Vela (~72° W, Figure 1a; see the review in [59]). Another upwelling occurs between Santa Marta and TNNP, typically during the peak of northern trade winds from December to March [55,56,59]. In both cases, the upwelled water is transported offshore by surface currents and added to the water carried by the CC toward the Central Caribbean [55,56,59] (Figure 1a). This putative barrier possibly limits larval dispersal from southwestern toward the northeast Caribbean coast of Colombia, affecting the genetic and phylogeographic structure of species associated with coral reefs and shallow rocky bottoms, whose marine ecosystems are absent for more than 300 km of coastline between Cabo de la Vela (La Guajira) and Tayrona National Natural Park, where upwelling occurs (Figure 1a).

Mollusks and fish are the most diverse and abundant species groups in the Colombian Caribbean [36,60] and serve as the biological models for investigating how the Magdalena River plume (MRP) and the combination of the absence of shallow rocky bottoms (ARB) with the almost permanent upwelling in La Guajira (PUG) affect the phylogeography of their populations. For this purpose, we chose three marine species with varying biological and ecological characteristics. These include the exposed rocky shore snails *Cittarium pica* (inhabits intertidal and shallow subtidal zones, exhibits high fecundity rate, external fecundity, and PLD < 6 days) [61,62], the reef fish *Acanthemblemaria rivasi* (inhabits invertebrate skeletal orifices embedded in rocks or corals in reef areas, exhibits egg parental care, low fecundity rate, external fecundity, and PLD < 22 days) [63,64], and another exposed rocky shore snail *Nerita tessellata* (inhabits splash and upper intertidal zones, exhibits high fecundity rate, external fecundity, and PLD > 60 days) [65,66,67]. It is worth noting that there is a recently proposed sister species for *A. rivasi* called *A. aceroi*. This specie was believed to be in Colombian waters from Santa Marta to Venezuela [68]. However, the separation of the two species was based on meristic and morphological aspects, and the molecular analysis mentioned by Hastings et al. [68] did not include Colombian samples to confirm *A. aceroi*’s presence in Colombia. Consequently, we considered *A. rivasi* as the species present in the Colombian Caribbean for our analysis. 

With this design, we followed aspect III of the multispecies approach proposed by Avise [4], which suggests comparative phylogeography studies to investigate whether the same factors that cause the spatial patterns of genetic structures can affect multiple codistributed taxa [4,69]. Therefore, we evaluated the following questions: Are the Magdalena River plume (MRP) and the combination of the absence of a rocky bottom and the almost permanent upwelling in La Guajira (ARB + PUG) physical barriers influencing the phylogeographic patterns of marine species with varying dispersal potential across the southern Caribbean Sea? What is the population model of each species? Furthermore, what is the level of congruence between the phylogeographic patterns of the three species?

**Figure 1 animals-13-02528-f001:**
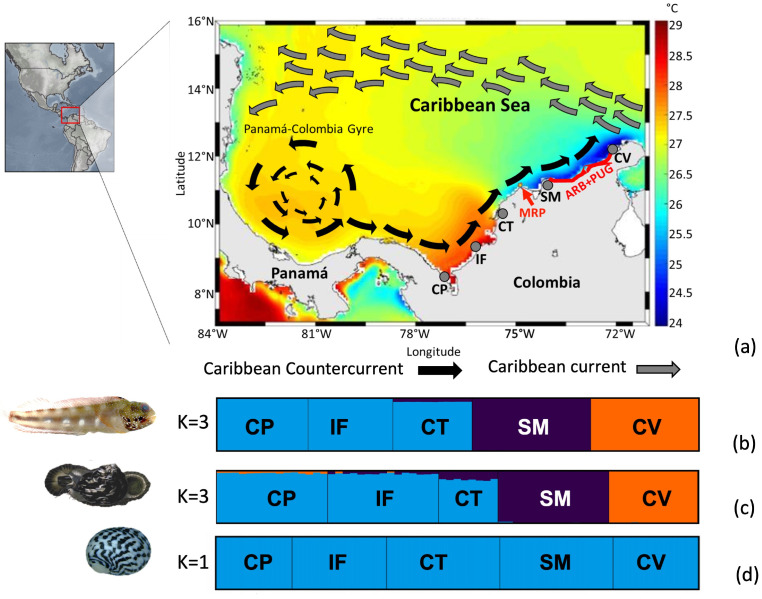
(**a**) Study area showing the five sampling localities, sea surface temperatures, and the main currents across the southern Caribbean Sea, Colombia sector. Localities: Capurganá (CP), Isla Fuerte (IF), Cartagena (CT), Santa Marta (SM), and Cabo de la Vela (CV). Gray arrow: Caribbean Current; black arrow: Caribbean Counter Current. The red lines indicate the putative barriers: the Magdalena River plume (MPR) and the combination of the absence of shallow rocky bottoms and the almost permanent upwelling in La Guajira (ARB + PUG). This map is taken and modified from CIOH (Pronóstico Climático del Caribe Colombiano No. 83, www.cioh.org.co). The scheme of currents is based on [48,70]. The bar graphs of the population ancestry coefficient from the STRUCTURE program indicate the most likely number of populations (*K*) for each species: (**b**) *Acanthemblemaria rivasi*, (**c**) *Cittarium pica*, and (**d**) *Nerita tessellata*.

## 2. Materials and Methods

### 2.1. Sampling

For each species, 95 specimens were collected across 5 marine sectors (1. Cabo de la Vela (12°27′29.3″ N, 71°39′59.6″ W–12°12′58.0″ N, 72°10′41.1″ W); 2. Santa Marta-Tayrona National Natural Park (11°18′51.6″ N, 74°11′39.1″ W–11°20′17.7″ N, 74°03′14.2″ W); 3. Cartagena-PNNN Corales del Rosario y San Bernardo (10°22′29.3″ N, 75°34′16.9″ W–10°15′22.2″ N, 75°37′04.5″ W–10°10′31.2″ N, 75°46′07.1″ W); 4. Isla Fuerte (9°23′15.4″ N, 76°10′23.6″ W), and 5. Capurganá-Sapzurro (8°38′30.6″ N, 77°20′45.0″ W–8°38′40.4″ N, 77°20′14.6″ W–8°39′40.6″ N, 77°21′40.3″ W; Figure 1a; Appendix A). The sampling was performed by scuba diving (*Acanthemblemaria rivasi*) and hand collection (*Cittarium pica* and *Nerita tessellata*). Foot (mollusks) and fin (fish) tissue samples were fixed with 96% ethanol. The final number of processed samples per location for each species is presented in Appendix A.

### 2.2. Laboratory Procedures

To obtain the SNP loci, double digest restriction site-associated DNA (ddRADseq) was developed using the library preparation protocol in [71]. The library preparation and sequencing of ddRADseq for 95 individuals were performed using the enzymes EcoRI and HpyCh4IV for *A. rivasi*, *Eco*RI and *Nla*III for *C. pica*, and EcoRI and HpyCh4IV for *N. tessellata*. The laboratory procedures were developed by the Australian Genome Research Facility. The libraries were selected according to size with targeting fragments 280–375 bp in size with Blue Pippin (Sage Science, Beverly, MA, USA), and PCR was amplified with indexed primers. Finally, the single-ended fragments were sequenced on an Illumina NextSeq500 with 150 cycles in high-output mode [72].

### 2.3. De Novo Assembly Reads

The processes to demultiplex reads by barcode and to remove reads with poor sequence quality or uncalled bases were developed in the program process_radtags in STACKS v2.0 [73,74]. Prior to the assembly of the RAD sequences, the optimal values of parameters *m* (minimum number of raw reads required to form a stack of a putative allele), *M* (number of mismatches allowed between putative alleles to merge them into a putative locus), and *n* (number of mismatches allowed between putative loci during the construction of the catalog) were obtained following the recommendations of [75]. First, the RADProc pipeline [76] was executed by selecting the 20 individuals (4 per location) with the highest number of reads. This program uses the same parameters defined in STACKS (*m*, *M*, and *n*) for de novo locus formation and catalog building. The output of the estimates allowed us to graph the number of loci, the number of polymorphic loci, and the number of SNPs of each combination of the parameter values [75], with the highest numbers observed when *M* = 6, *m* = 3, and *n* = 2 for *A. rivasi*, *M* = 5, *m* = 3, and *n* = 2 for *C. pica*, and *M* = 4, *m* = 3, and *n* = 2 for *N. tessellata*. These values were established to assemble the rad loci with the *denovo_map* program in STACKS v.2.0 [74].

Without a reference genome, RAD-seq loci were assembled de novo using the *denovo_map* program in STACKS [74]. The final filtered genotype file was created by applying the following filters: a minimum percentage of individuals in the populations required to process a locus (R = 0.8) equivalent to 80% and a minor allele frequency (MAF) (*min_maf* = 0.02). The negative effects of missing data were minimized when calculating frequency-based genetic distance parameters. SNPs were also screened for allele coverage, with any SNPs displaying a local and global minor allele frequency (MAF) threshold of less than 1% removed from the dataset. In cases where multiple SNPs were found within the same read, only one locus was retained (*write_single_snp*) in order to avoid statistical bias from the physical linkage. Individuals who exceeded the missing data percentage (less than 5%) were removed from the final file. After the final filtering, 85, 65, and 51 individuals of *A. rivasi*, *C. pica*, and *N. tessellata*, respectively, remained. The VCF file was also converted into other formats using PGDSpider v2.1 software [77].

### 2.4. Analysis of Data

#### 2.4.1. Genetic and Phylogeographic Structure

Genetic diversity indices were estimated by the number of alleles, the effective number of alleles, observed heterozygosity (*H*o), heterozygosity within populations (*H*s), and inbreeding coefficient (*G*_IS_), which were generated with GenoDive [78]. As theisstudy aims to determine levels of variation at the intra-locality level on biogeographic barriers, the results of these parameters are presented in Appendix A.

We determined the spatial genetic structure of each species to identify the probable number of populations (*K*) using the Bayesian approach with the program STRUCTURE v.2.3.3 [79]. Each run included 200,000 burn-in iterations, followed by 800,000 MCMC iterations used to calculate a probable K between 1 and *n* + 1 populations in each case, with 7 runs for each simulated K value. The admixture and allele frequency correlated models were assumed here. We determined the number of most likely populations with the methods proposed by [80,81,82] using the STRUCTURE SELECTOR program [70], which includes the CLUMPAK program [83], to combine and visualize the STRUCTURE results. Individuals within populations were ordered according to sampling localities. The STRUCTURE SELECTOR program allowed us to choose the K method by Raj et al. [81] for large SNP datasets and measures like LnP(K), Delta K, MaxMedK, MaxMeanK, MedMedK, and MedMeanK [80,81,82]. LnP(K) is the log-posterior probability of a model with K populations, and Delta K is the difference between the log-posterior probabilities of two models with different numbers of populations [80]. The Max, Med, and Mean refer to the maximum, median, and mean of the log-posterior probabilities of all models with a given number of populations. MedMedK and MedMeanK are the median and mean of the Delta K values for all models with a given number of populations [82]. Higher values of these measures indicate a higher likelihood of a model with more populations [70]. Additionally, a principal component analysis (PCA) was performed for each species with the *dudi.pca* function of the ade4 package [84] in R.

We performed estimates of Φ_ST_ to determine genetic differentiation between pairs of localities with STACKS [74] using the “populations” module with the “–*fstats*” indicator. In addition, we performed an analysis of molecular variance (AMOVA) to determine the effects of putative barriers on the phylogeographic pattern of each species. This evaluated several levels of grouping according to the localities on each side of the putative barriers. 1. The effect of the absence of shallow rocky bottoms for more than 300 km of coastline between Cabo de La Vela and Santa Marta, as well as the almost permanent upwelling of La Guajira (ARB + PUG). 2. The effect of the Magdalena River plume. 3. A third analysis was performed for the species that presented substructuring according to the results of the Bayesian analysis of STRUCTURE (only for *C. pica* and *A. rivasi*). AMOVAs were performed with the *poppr.amova* function in the poppr package in R [85].

A phylogeographic pattern analysis was performed from phylogenetic analyses based on consensus sequences for all concatenated individuals. The files were generated using the STACKS population program with the –*phylip_var* function based on 477,072 SNPs for *A. rivasi*, 367,676 SNPs for *C. pica*, and 229,422 SNPs for *N. tessellata*. The generated matrices were used to perform maximum likelihood (ML) analyses with the program IQTREE 1.3.10 [86] with 2 independent runs, including model selection with SNP verification bias correction, 1000 ultrafast bootstrap replicates, and 10,000 iterations. The SNP-based sequence evolution model was selected by IQTREE following the Bayesian information criterion, with TVM + ASC + G4 for *A. rivasi*, TVMe + ASC + G4 for *C. pica*, and TVMe + ASC + G4 for *N. tessellata*. 

#### 2.4.2. Population Model

We determined the population model of each species (open, isolated, or hierarchical populations) with the Mantel test to correlate the genetic (Φ_ST_) and geographic (km) distance matrices using the *mantel* function in the R package vegan 2.5.7 [87]. Two geographic distances were calculated in Google Earth: one directly measured the distance between locations (linear), and the other measured along the coastline. We developed the analyses without any previous manipulation of data [88]. Additionally, a Mantel correlogram analysis was performed, dividing the geographic distances into submatrices of distance class marks [88]. We performed this to determine the false positives of isolation by distance when the populations were genetically structured because of biogeographic barriers [89]. Determining the false positives of IBD to understand population differentiation accurately is crucial. For example, while IBD suggests that genetic differences between populations increase as they become more geographically distant, biogeographic barriers can cause genetic differentiation even at shorter distances. Therefore, an overestimation of the impact of geographic distance can occur, which is necessary to distinguish between the effects of geographic distance and biogeographic barriers.

#### 2.4.3. Phylogeographic Concordance between Species

A phylogeographic concordance analysis among species was performed by comparing the topology of the dendrograms constructed with the pairwise Φ_ST_ matrices calculated between localities. In this sense, the Kendall procedure [90] was used to compare dendrogram topology according to tip category names using a concordance factor (CF). Each tip was labeled according to the localities when the dendrograms were created. The CF measure can estimate values between 0 and 1. It is 1 when the concordance is complete. For this measure, comparisons were made between *A. rivasi* and *C. pica*, *A. rivasi* and *N. essellate*, and *C. pica* and *N. essellate*. The procedure was performed with the *treeConcordance* function in the R package treeSpace [91], entering the matrix in the Newick format of each dendrogram. Another estimator employed was the congruence between the distance matrices test (CADM) [92], which was used to determine topological and genetic congruence. This test evaluates the hypothesis of complete incongruence between species trees, corresponding to phylogenies with different topologies or branch lengths [92]. Kendall’s *W* statistic was used to determine the level of congruence between species, which has values between 0 and 1, with 0 indicating no congruence and 1 being complete topological or genetic congruence of the trees. The topological and genetic congruence tests used the Φ_ST_ matrices calculated between pairs of locations.

The CADM test was run for each congruence type and between pairs of species with the *CADM.global* function in the R package Ape [93]. Only for the topological congruence test was the *cophenetic.phylo* function in the R package Ape [93] previously executed. This function calculates the cophenetic distances between the tips of a phylogenetic tree or dendrogram. The cophenetic distance is the shortest distance between two tips in the tree or dendrogram. The trees or dendrogram have the same topology if the cophenetic distances are significantly similar. In this case, the significance value was estimated from 99,999 permutations. As a complement, the Mantel test was performed between the triangular matrices of the paired Φ_ST_ values of the codistributed species with the *mantel* function in the R vegan 2.5.7 package [87]. Congruence between the matrices is indicated by *r* values close to 1 and *p* < 0.05. The significance value was estimated from 99,999 permutations.

## 3. Results

We obtained 10,122,277 reads for 85 individuals of *Acanthemblemaria rivasi*, 8,764,529 reads for 65 individuals of *Cittarium pica*, and 3,385,649 reads for 51 individuals of *Nerita essellate*. The average number of usable polymorphic SNP loci per sample varied: 66,325 were determined for *A. rivasi*, 55,151 for *C. pica*, and 21,333 for *N. essellate*. A summary of the genetic diversity indices is presented in Appendix A. 

### 3.1. Genetic and Phylogeographic Structure

For *Acanthemblemaria rivasi*, the pairwise Φ_ST_ comparisons were significant between all pairs of localities. A considerable genetic differentiation between samples from Cabo de la Vela and the other localities was determined (Φ_ST_ > 0.290 values; *p* < 0.05), with a greater magnitude observed for the most distant location (Capurganá; Φ_ST_ = 0.345; *p* < 0.05). Samples from Santa Marta also showed high genetic differentiation compared to the rest of the localities (Φ_ST_ > 0.262; *p* < 0.05), while those collected in Cartagena, Isla Fuerte, and Capurganá had moderate differentiation among them (Φ_ST_ < 0.097; Table 1). The results of the Bayesian analysis provided variable results among the estimates of the probable number of populations. For example, the ∆*K* estimate provided *K* = 2. According to the bar graph, the first population is distributed in Cabo de la Vela, and the second in Santa Marta, Cartagena, Isla Fuerte, and Capurganá (Appendix A). The method by Raj et al. [81] and the MedMedK and MedMeanK estimates indicated *K* = 3, with the first population located at Cabo de la Vela again, the second at Santa Marta, and the third across Cartagena, Isla Fuerte, and Capurganá. For the LnP(*K*), MaxMedK, and MaxMeanK estimates, *K* was 5. These estimates proposed the same first three populations as with *K* = 3, but with the fourth (higher proportion) and fifth populations (in a minimum proportion) codistributed mainly between Cartagena and Isla Fuerte (Appendix A). Principal component analysis (PCA) and the network tree showed three clusters, which were congruent with the *K* = 3 of the Bayesian analysis and the levels of differentiation based on Φ_ST_ (Figure 2a).

In the case of *C. pica*, the samples from Cabo de la Vela were also differentiated from the rest of the localities. Although this genetic differentiation was moderate (Φ_ST_ > 0.130; *p <* 0.05), the highest value was observed for the neighboring locality of Santa Marta (Φ_ST_ = 0.175; *p* < 0.05). The localities of Cartagena, Isla Fuerte, and Capurganá presented a lower level of differentiation between their samples (Φ_ST_ < 0.013; *p <* 0.05; Table 1). Concerning the Bayesian analysis, the LnP(*K*), MedMedK, MedMeanK, MaxMedK, and MaxMeanK estimates and the method by Raj et al. [81] calculated *K* = 3, with one population located in Cabo de la Vela, the second in Santa Marta, a minimal proportion in Cartagena (<5%), and the third codistributed across Cartagena, Isla Fuerte, and Capurganá (Figure 1c). Under this population number proposal, a minimal proportion of the population located in Cabo de la Vela is present in Isla Fuerte and Capurganá. The ∆*K* estimated *K* = 5, showing the same results as in *K* = 3, but determining a fourth and fifth cluster in minimal proportions (<3%) and codistributed between Isla Fuerte and Capurganá (Appendix A). The PCA is congruent with the proposed *K* = 3 and the Φ_ST_ results (Figure 2b).

*Nerita tessellata* presented slight genetic differentiation among all localities, with an average value of Φ_ST_ = 0.006, which was the highest but had a subtle differentiation compared to Cabo de la Vela and the rest of the localities (Φ_ST_ < 0.009; Table 1). The MedMedK, MedMeanK, MaxMedK, and MaxMeanK estimates, and the method by Raj et al. [81] determined a *K* = 1, which is distributed at all localities (Figure 1d). For the cases of ∆*K* and LnP(*K*), *K* values equal to 3 and 2, respectively, were proposed. The two clusters are distributed from Cabo de la Vela to Capurganá, one with a higher proportion than the other. The third cluster proposed by LnP*(K*) is distributed in a minimal proportion in all localities except Cabo de la Vela (Appendix A). However, the PCA and network tree agree with the *K* = 1 proposal (Figure 2c).

### 3.2. Identification of Phylogeographic Breaks

The genetic differentiation analysis by AMOVA presented a level of phylogeographic structuring in all species (Φ_ST_ > 0.076, *p* < 0.05), except *N. tessellata* (Φ_ST_ = 0.001–0.002, *p* > 0.05), which did not present substructuring (Table 2). For *A. rivasi*, the leading causes of the phylogeographic pattern were two putative barriers: 1. the combination of the absence of shallow rocky bottoms and the almost permanent upwelling (ARB + PUG) located between Cabo de la Vela and Santa Marta (AMOVA: Φ_CT_ = 0.406, *p* < 0.05) and 2. the Magdalena River plume (MRP; Φ_CT_ = 0.420, *p* < 0.05). These two barriers delimit the three populations that were identified as *A. rivasi*. Under this last scenario, a third AMOVA was performed assuming K = 3, determining a high genetic differentiation (Φ_CT_ = 0.495, *p* < 0.05; Table 2). The maximum likelihood (ML) tree for *A. rivasi* was concordant with the AMOVA when *K* = 3 was assumed, confirming the effects of the two putative barriers (Figure 3a). The three clades are observed with 100% bootstrap support. The same as in the Bayesian analysis when *K* = 3, one clade is constituted by the Cabo de la Vela samples, the second by those from Santa Marta (100% bootstrap), and the third by the Cartagena + Isla Fuerte + Capurganá samples (100%).

Concerning *C. pica*, when the ARB + PUG barrier was assumed, substantial genetic differentiation was observed between the samples from both sides (Φ_CT_ = 0.224, *p* < 0.05) compared to the MRP barrier, which showed moderate genetic differentiation (Φ_CT_ = 0.076, *p* < 0.05). Furthermore, when the analysis was performed assuming *K* = 3, the differentiation was significant (Φ_CT_ = 0.174, *p* < 0.05; Table 2). However, the phylogenetic analysis was concordant with the ARB + PUG barrier (100% bootstrap). Only two clades are configured, one on each side of the barrier (clade 1. Cabo de la Vela and clade 2. Santa Marta + Cartagena + Isla Fuerte + Capurganá). Interestingly, clade 2 shows that the Santa Marta samples are more closely related but separated from the rest of the localities, with 53% bootstrap support (Figure 3b). With respect to *N. tessellata*, no evidence of phylogeographic breaks was observed (Figure 3c).

### 3.3. Population Model

Only *A. rivasi* showed a significant correlation between the genetic distance and linear and coastline distances (*r*_m coastline_ = 0.623, *p* = 0.0253; *r*_m linear_ = 0.774, *p* = 0.0118; Figure 4a), but no significant correlations were found for *C. pica* (*r*_m coastline_ = 0.441, *p* = 0.2059; *r*_m linear_ = 0.616, *p* = 0.0571; Figure 4b) and *N. tessellata* (*r*_m coastline_ = 0.422, *p* = 0.227; *r*_m linear_ = 0.580, *p* = 0.0831; Figure 4c) was not significant. However, the Mantel correlogram indicated that *A. rivasi* and *C. pica* fit the hierarchical population model (Appendix A).

### 3.4. Phylogeographic Concordance between Species

A concordance factor of 0.6 was determined when comparing the topologies of the dendrograms of *A. rivasi*-*N. tessellata* and *C. pica*-*N. tessellata* and 0.5 for *A. rivasi*-*C. pica*. The global CADM test showed significant topological (*Wt* = 0.847, *X*^2^ = 22.86, *p* = 0.012) and genetic (*Wg* = 0.852, *X*^2^ = 23, *p* = 0.011) differences among the three species. However, in pairwise comparisons, only *A. rivasi*-*C. pica* showed significant topological (*Wt* = 0.937, *X*^2^ = 16.88, *p* = 0.046) and genetic (*Wg* = 0.915, *X*^2^ = 16.42, *p* = 0.048) correlations, as well as a significant correlation between genetic distance matrices (*r*_m_ = 0.7615, *p* = 0.001). *N. tessellata* only presented genetic congruence with *A. rivasi* (*Wg* = 0.915, *X*^2^ = 16.47, *p* = 0.043; Appendix A).

## 4. Discussion

### 4.1. Genetic and Phylogeographic Structure

Genetic structuring analyses based on paired Φ_ST_, Bayesian analysis, PCA, AMOVA, and ML trees determined the most likely number of clusters for the three marine species sampled along the southern Caribbean Sea (Colombia sector). However, only *N. tessellata* presented one population (*K* = 1), and its high dispersal potential reflected in a larval lifespan of more than 60 days [65,66,67] may be the primary explanation for the absence of genetic structuring. The overall Φ_ST_ value was low and coincides with those determined for other marine species associated with the southern Caribbean ecoregion, which were estimated with various molecular markers (mitochondrial genes, SNPs, and microsatellite loci), mainly in the fish *Stegastes partitus* [46], *Lutjanus synagris* [47], *Caranx hippos* [49] and *Micropogonias furnieri* [94], the rock boring urchin *Echinometra lucunter lucunter* [51], the southern white shrimp *Litopenaeus schmitti* [95], and the Caribbean sharpnose shark *Rhizoprionodon porosus* [96]. Although some Φ_ST_ values were significant, the levels of genetic differentiation between localities were too low to prove any phylogeographic break (*F*_ST_ < 0.05), and the high dispersal potential of these species may be a reason for the lack of structure (PLD > 10 days). Other studies that extended the sampling coverage to other regions of the Great Caribbean documented that species with high PLD did not show a phylogeographic pattern, i.e., *Echinolittorina ziczac* and *Cenchritis muricatus* [48], *Panulirus argus* [97] and some other species had a panmictic population pattern, such as *Sparisoma viride* [29], *Clibanarius tricolor* [30], and *Sparisoma aurofrenatum* [53]. Overall, most likely the length of the larval period of *N. tessellata* defined the phylogeographic structure with a single population. This species also exhibits high genetic connectivity among localities due to the hydrographic patterns of the Caribbean Current and the Colombia Counter Current (CCC), which could generate larval dispersal along the southern Caribbean despite the presence of the barriers evaluated (Figure 1a). 

Concerning *A. rivasi* and *C. pica*, pairwise Φ_ST_, Bayesian analysis, and PCA allowed the identification of three populations (*K* = 3), which were confirmed with AMOVA. The delimitation of the three populations is influenced by the putative barriers evaluated. For example, based on the Φ_CT_ statistic of the AMOVA, it was possible to measure the level of genetic differentiation between the samples classified on either side of the barriers. For both species, the first population was in Cabo de la Vela and the second was in Santa Marta. The populations are separated by the barrier defined as a combined effect of the absence of the rocky bottom at a distance greater than 300 km from the coastline (ARB) and the almost permanent upwelling in La Guajira (ARB + PUG). This barrier significantly affects *A. rivasi* (Φ_CT_ = 0.406) and *C. pica* (Φ_CT_ = 0.224) and somewhat effects *N. tessellata* (Φ_CT_ = 0.005). Finally, the analysis determined a third population among the localities of Cartagena, Isla Fuerte, and Capurganá, which are separated from the second by the barrier of the Magdalena River plume (MRP). This barrier significantly affects *A. rivasi* (Φ_CT_ = 0.420) and, to a lesser degree, *C. pica* (Φ_CT_ = 0.076); for the latter, it is a barrier permeable to gene flow. *Cittarium pica* presents a PLD of less than six days [61,62]. This time may be sufficient for larvae produced in Cartagena to be transported to Santa Marta via the CCC [55,56,58]. The CCC may also be responsible for maintaining high genetic connectivity between Capurganá and Cartagena, where the third populations of *C. pica* and *A. rivasi* are found. It is assumed that the larvae of both species produced in the reef systems of the Cartagena (Tierrabomba, Barú Island, Rosario Islands, and San Bernardo archipelago), Isla Fuerte, and Capurganá sectors are transported by CCC [58]. This current has been proposed as the oceanographic factor that facilitates the exchange of genetic information among populations of several marine species that do not exhibit genetic structure in the southwestern Caribbean, i.e., *S. partitus* [46], *L. synagris* [47], *L. schmitti* [95], *R. porosus* [98], *C. hippos* [49], *Orbicella faveolata* [45], *P. notialis* [52], *Mugil liza* [99], *Acropora palmata* and *A. cervicornis* [54], and *Micropogonias furnieri* [94].

The phylogenetic analysis confirmed the previous phylogeographic proposal for *A. rivasi* and *C. pica*. For *C. pica*, only two clades were configured in the ML tree, one on each side of the ARB + PUG barrier. Nevertheless, clade 2 shows that the samples from Santa Marta are more closely related to the rest of the localities. This was confirmed by the Bayesian analysis with a *K* = 3, showing that a small portion of the population from Santa Marta is also present in Cartagena. All this suggests that *C. pica* shows evidence of the effect of a single biogeographic barrier (ARB + PUG). In contrast, for *A. rivasi*, the simultaneous impact of the two barriers evaluated (ARB + PUG and MRP) is observed. These results are interesting because the action of the two barriers was identified for a reef species, such as *A. rivasi,* less than 400 km from the coastline. Interestingly, species of the genus *Acanthemblemaria* have a proposed PLD of 22 to 25 days [64]. If this is true for *A. rivasi*, under this dispersal scenario, it would be affected by only one of the two barriers, as in the case of *C. pica*. It is possible that this fish could have a PLD of fewer than 10 days, which must be investigated. However, we believe that another factor might explain the phylogeographic pattern of *A. rivasi.* For example, the *Acanthemblemaria* fish are characterized by parental care of eggs by males, low fecundity, and inhabiting invertebrate skeletal orifices embedded in rocks or corals in reef areas with high wave energy [63,64,100]. In addition, these species show larval retention near coral reefs [101,102]. The reproductive and ecological characteristics mentioned above could generate high larval retention and high biological recruitment in the local populations of Cabo de la Vela and Santa Marta which, when added to the effects of ARB + PUG and MRP barriers, define the high level of genetic structuring observed. 

The investigation undertaken has yielded evidence of the existence of another *Acanthemblemaria* species in the waters of Colombia extending from Santa Marta to Cabo de la Vela and is likely to be *A.* cf *aceroi*. The outcomes indicate a significant genetic differentiation between samples procured on opposite sides of the mouth of the Magdalena River, specifically from Santa Marta–Cabo de la Vela in comparison to Cartagena–Isla Fuerte-Capurganá (AMOVA: Φ_CT_ = 0.495, *p* < 0.05). The existence of *A. rivasi* has likewise been ascertained between Costa Rica and Cartagena. This argument was confirmed by the phylogenetic and divergence analysis based on sequences of the COI gene, which revealed a divergence surpassing 9% among the samples composed of Cabo de la Vela-Santa Marta and Cartagena-Isla Fuerte-Capurganá. Also, a divergence was determined to surpass 9% between Cabo de la Vela-Santa Marta samples and sequences Panamanian of *A. rivasi* deposited in GenBank (Appendix A). However, there were no divergences found between Panama and Cartagena-Isla Fuerte-Capurganá sequences. These findings highlight the need for further research that includes samples from Venezuela to determine the distribution and prevalence of *A. aceroi* in Colombian waters between Santa Marta and Cabo de la Vela.

### 4.2. Identification of Phylogeographic Breaks, Population Models, and Phylogeographic Concordance

The phylogeographic analysis of the three marine species determined the action of two putative barriers across the southern Caribbean (Colombia sector). For the first time, evidence is provided of a phylogeographic break caused by the Magdalena River Plume (MRP), mainly for *A. rivasi*. The Magdalena River annually delivers 142 × 10^6^ tons year^−1^ of sediment to the Caribbean Sea. The MRP extends 6.5 km offshore and is characterized by high turbidity (178.6 ± 78.7 mg L^−1^), comparable to estimates for the Amazon and Yangtze rivers [103]. Furthermore, MRP dilutes the salinity from 36 in the open ocean to 28.4 ± 0.4 at 6.5 km offshore, and to 10.8 ± 3.4 at the mouth of the Magdalena River [104]. These physicochemical conditions and their influence on the Caribbean Sea suggest that the PMR must act as a biogeographic barrier for marine species that are limited in their dispersal, mainly those whose pelagic larvae are unable to survive when attempting to cross this area. Perhaps they cannot tolerate the decrease in salinity and higher temperatures (≥2 °C) in the marine waters off MRP, as well as the high turbidity of the waters, which are possible factors that regulate the success of biological dispersal. The MRP likely operates as a filter for *A. rivasi* larvae that attempt to cross it through the action of the CCC. However, the possibility that populations formed on either side of the plume have adapted to the different environmental conditions is not excluded. For example, larvae that cross the MRP may be selected against because the marine waters on the western side are warmer and less salty than the eastern side [105]. 

The MRP was proposed as a barrier across the southern Caribbean [41]. However, many studies were inconclusive because some of them did not consider localities in Colombia [39,40,42,43,44], and when it was possible, they only sampled one side of the MRP [41,45]. The hypothesis of the MRP as a barrier was rejected when studies included both sides of the MRP and selected species with pelagic larvae enduring longer than 12 days (*S. partitus*, *L. synagris*, *L. schmitti*, *C. hippos*, *M. incilis*, *P. notialis*, *M. liza*, *M. furnieri*, *A. cervicornis*, and *A. palmata*). Nevertheless, marine species in some parts of the world are separated by phylogeographic breaks created by large river plumes. For instance, the Amazon River plume has been studied for its impact on the genetics and biogeography of reef fish. These studies have shown that the fish fauna in the Caribbean Sea are distinct from Brazil, and certain species exhibit genetic differences between populations in both areas (i.e., *Chromis multilineata*, *Halichoeres* spp., *Lutjanus synagris*) [12,32,106,107]. Another phylogeographic break caused by the Yangtze River plume has also been documented (i.e., for the limpet *Cellana toreuma*) [24].

The other finding was the effect caused by the absence of a rocky coastline for more than 300 km between Cabo de la Vela and the TNNP sector, which operates on *A. rivasi* and *C. pica*. A rocky coastline constitutes a specific habitat for both species, which show heterogeneous distributions in the southern Caribbean [51,108]. However, the absence of a rocky coastline in this 300 km sector coincides with two upwelling areas. One is almost permanent in La Guajira, with strong upwelling during December–May and a weak effect during June–August when the temperature reaches a minimum of 24 °C (~72° W, Figure 1a; see the review in [59]). The other upwelling develops between Santa Marta and TNNP and is usually seasonal (December to March), with a minimum of 24 °C [55,56,59]. In both cases, surface currents transport upwelled water offshore [55,56,59], which is then added to water transported by the Caribbean Current toward the Central Caribbean [55,59] (Figure 1a). This oceanographic feature may be responsible for transporting larvae offshore [59] and regulating the genetic exchange between the Cabo de la Vela and Santa Marta localities. Therefore, it could be a combined effect that generates the phylogeographic break in the two species. In this study, we documented the ARB + PUG barrier for first time [51,53] and is interesting because the two evaluated species display contrasting biological and ecological aspects [63,109], which could be a signature of the generalized effect of the barrier on the communities of marine organisms dependent on the rocky bottoms across the southern Caribbean aspect III [4]. A similar situation occurs in Norway with the rocky shore fish *Symphodus melops* [21] and in southeastern Australia with the barnacle *Catomerus polymerus* and the limpet *Cellana tramoserica* [5], where these marine species associated with the rocky shore exhibit breaks in genetic connectivity due to a biogeographic barrier formed by sandy coastlines.

Some investigations have explained how upwelling zones affect the genetic and phylogeographic structures of marine species. For example, upwelling events at Cape Blanco (Oregon) and Cape Mendocino (California) affect the genetic structure of the barnacle *Balanus glandula* [110] and five species in the rocky intertidal community [27], respectively. Along the southeastern Pacific coast, the gastropod *Crepipatella dilatata* [111] and the beach isopod *Excirolana hirsuticauda* [112] exhibit a biogeographic break at 32° S, a transition area characterized by upwelling. In addition, the fish *Pomatomus saltatrix* exhibits this phenomenon in the upwelling area of the Benguela Current [113], and the fish *Sebastes thompsoni* occurs between two sectors of the East Sea, which may be related to current patterns, such as eddies and upwelling [25]. 

One of the assumptions in population genetics is that genetically structured species fit the isolation by distance (IBD) model, in which samples from nearby locations are less genetically different than those farther apart [114]. However, highly structured species may exhibit false positives when testing the IBD model for spatial autocorrelation between genetic and geographic distances [88,89]. Some species that experienced abrupt separations in the past are evidenced by high levels of genetic divergence in the present. Thus, Mantel test analyses and Mantel correlograms performed on *A. rivasi* and *C. pica* determined that neither of them fit the IBD model. In contrast, these analyses demonstrated that they fit the hierarchical population model [89]. This model coincides with those species that show an abrupt genetic change corresponding to a biogeographic barrier in a geographical area [1,89,115]. 

Species that do not fit the IBD population model have been investigated across the geographical area studied, principally those species with high PLD values and low levels of genetic differentiation that were estimated from microsatellite loci, as in the cases of *S. partitus* [46], *L. synagris* [47], *M. incilis* [50], *E. lucunter lucunter* [51], *Mugil liza* [99], and *C. hippos* with mitochondrial genes [49]. However, the literature review identified that the shrimp *P. notialis* is the only species that fits the IBD model across the southern Caribbean [52]. For this species, three populations were determined, with the samples from La Guajira being the most genetically different from the rest of the localities in Colombia.

Phylogeographic concordance factors allow us to quantify the degree of phylogeographic congruence among species to determine which species share a phylogeographic pattern [4,116,117]. However, most work concludes that not all codistributed species exhibit identical phylogenies due to incongruence in tree topology and clade divergence times. This argument is posited because each species responds differentially to the factors responsible for the phylogeographic break and the biological characteristics of the species involved [5,69,118]. In this study, *A. rivasi* and *C. pica* presented phylogeographic congruence, coinciding in presenting phylogeographic breaks to the putative barriers evaluated but with a differential response. For example, *A. rivasi* showed two phylogeographic breaks caused by the ARB + UPG and MRP barriers, while *C. pica* was only affected by ARB + UPG. Additionally, the biological and genetic attributes of both species are likely responsible for the level of congruence observed, including a low dispersal potential attributed to a PLD of a few (6 days in *C. pica*) to several days (22 days in *A. rivasi*); egg parental care (*A. rivasi*) and low fecundity (*A. rivasi*); high Φ_ST_ values (> 0.08); *K* = 3 (*A. rivasi*, *C. pica*); and two (*C. pica*) to three clades (*A. rivasi*) delineated by the phylogenetic analyses.

### 4.3. Conservation Aspects

The spatial pattern for *A. rivasi* and *C. pica* coincides with the zoogeographic subareas proposed by Díaz [36] as a result of a study of the biogeography of marine gastropods in the southern Caribbean. This author proposed five subareas, three of which are distributed in the marine sector of Colombia. The first is located at the mouth of the Magdalena River to Costa Rica (Isthmian), which contains the Colombian localities of Cartagena, Isla Fuerte, and Capurganá. The second includes the marine sector at the eastern side of the mouth of the Magdalena River to the eastern side of Tayrona Natural National Park (Samarian), where the third subarea begins and reaches the Paraguaná Peninsula in Venezuela (Goajira). This subarea includes the Cabo de la Vela location. Overall, these three subareas are characterized by differences in the size of the continental shelf, the predominant types of bottoms, the conditions of calm or agitated waters, the transparency, temperature, and salinity of the water masses, and the different types of habitats that they contain; see details in [36]. 

The physical, chemical, and geological conditions of the three subareas exert substantial effects on the coastal marine species of the Colombian Caribbean, which allowed us to observe a typical pattern of genetic differentiation between the samples collected in the Goajira subarea and the other subareas, i.e., *C. mapale* [35], *E. lucunter* [51], *P. notialis* [52], *R. porosus* [98], *Sciades proops*, and *Melongena melongena* [94]. At La Guajira, populations of each species should be considered as genetic management units (GMUs) to prioritize implementing conservation and fisheries management measures [119], mainly for those that are exploited by fishing activities or those that are part of fragile marine ecosystems, such as mangroves, coral reefs, and seagrass. Although two marine protected areas are established in La Guajira (Bahía Portete—Kaurrele National Natural Park and Los Flamencos Sanctuary) they are not sufficient for conservation.

On the other hand, additional *C. pica* and *A. rivasi* populations were identified in the Samarian subarea, which should also be treated as a second GMU. This sector has the advantage of including protected marine areas, such as Tayrona Park and Salamanca Island Road Park, where the main factors affecting species and ecosystem conservation, such as fishing and tourism activities, are regulated. The population associated with the Isthmian subarea is the largest and should be considered the third GMU for both species. Although the protected marine areas of Corales del Rosario, the San Bernardo Islands, and Acandí are established there, the challenge for the fishery and environmental authorities is to improve the regulation of multiple human activities that affect the conservation of species and ecosystems outside these protected areas.

Finally, additional studies should be conducted to investigate the phylogeographic patterns of each species assuming all areas of its distribution. The results will serve as a basis for implementing a multinational approach to developing conservation strategies to which the countries that host the species will be committed. 

## 5. Conclusions

This study demonstrated for the first time the phylogeographic break caused by the MRP for *A. rivasi,* the combined effect caused by the absence of a rocky bottom along more than 300 km of coastline, and the almost permanent upwelling in La Guajira (ARB + PUG), which operates on *A. rivasi* and *C. pica*. Three populations (*K* = 3) were identified as *A. rivasi* and *C. pica*, while *N. tessellata* presented one population and exhibited the panmixia model (*K* = 1). Only *A. rivasi* and *C. pica* showed phylogeographic congruence. This could be due to the oceanographic and environmental conditions of the Colombian Caribbean generating discordance over species with short larval phases. However, other biological and ecological traits of these species are crucial for understanding the phylogeographic structure (e.g., egg parental care, high or low fecundity, species interactions, and the effect of the environment on the ontogeny of the organisms). They should be considered for further research.

The above demonstrates the importance of further investigating the effects of the MRP and ARB + PUG barriers on other species inhabiting the rocky shores and coral reefs of the southern Caribbean. Ideally, that research should use a multispecies approach that includes species from different taxonomic groups to test whether the effects of the barrier are widespread throughout the marine community associated with this ecosystem [4,120]. In addition, research should also be based on the multilocus approach [4] to evaluate whether barriers have generalized genomic effects on marine species.

## Figures and Tables

**Figure 2 animals-13-02528-f002:**
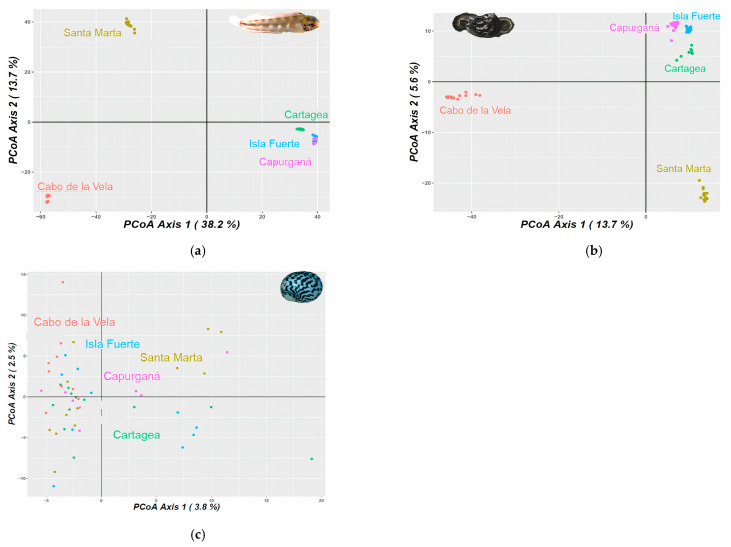
Principal component analysis (PCA) for (**a**) *Acanthemblemaria rivasi*, (**b**) *Cittarium pica*, and (**c**) *Nerita tessellata* across the southern Caribbean Sea, Colombia sector. The percentage of variation is exhibited by two axes.

**Figure 3 animals-13-02528-f003:**
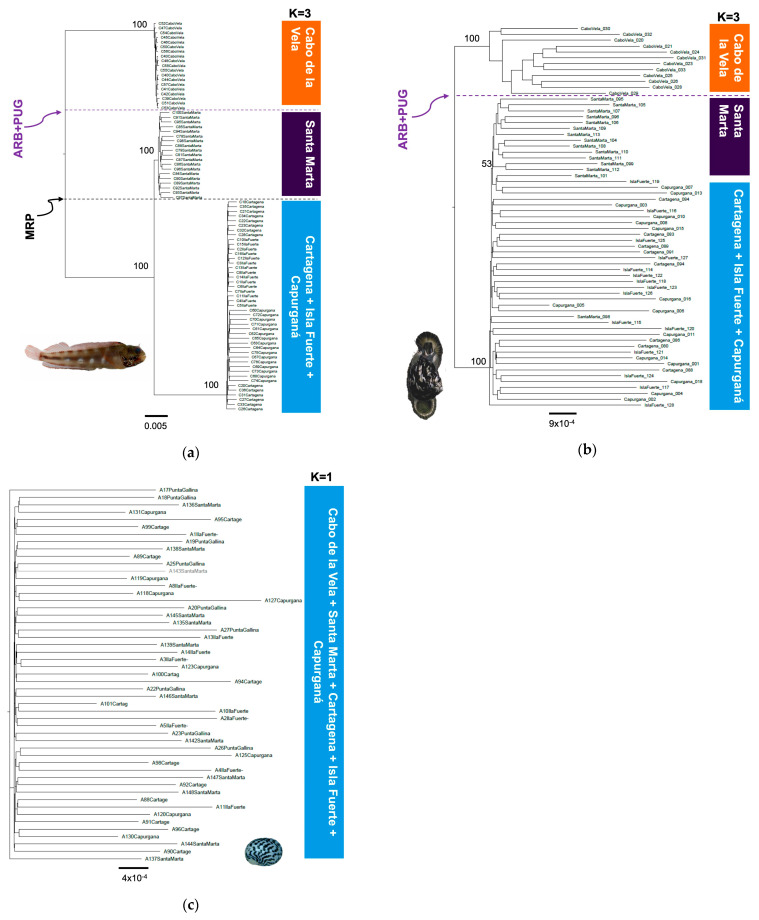
Phylogenetic trees constructed with the maximum likelihood method with the SNP matrix and the representation of the results of the STRUCTURE analysis indicating the most likely *K* number and related localities in the southern Caribbean (Colombia sector) for (**a**) *A. rivasi*, (**b**) *C. pica*, and (**c**) *N. tessellata*. The dotted lines indicate putative barriers (MRP: Magdalena River plume; ARB + PUG: the combination of the absence of a rocky bottom and the almost permanent upwelling in La Guajira).

**Figure 4 animals-13-02528-f004:**
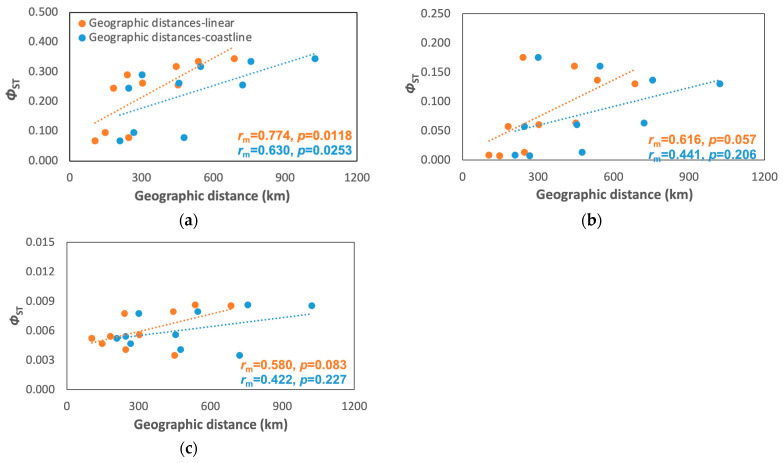
Mantel test plots between geographic distances (linear and coastline) and Φ_ST_ for (**a**) *A. rivasi*, (**b**) *C. pica*, and (**c**) *N. tessellata*. *r*_m_ = correlation coefficient from the Mantel test.

**Table 1 animals-13-02528-t001:** Pairwise comparison of Φ_ST_ between the five localities sampled across the southern Caribbean (Colombia sector) for the species *Acanthemblemaria rivasi*, *Cittarium pica*, and *Nerita tessellata*. *: values showed a significance level of *p* < 0.005. ^n.s.^: not significant.

** *Acanthemblemaria rivasi* **				
	Santa Marta	Cartagena	Isla Fuerte	Capurganá
Cabo de la Vela	0.290 *	0.318 *	0.336 *	0.345 *
Santa Marta		0.246 *	0.262 *	0.257 *
Cartagena			0.069 *	0.080 *
Isla Fuerte				0.097 *
** *Cittarium pica* **				
Cabo de la Vela	0.175 *	0.160 *	0.136 *	0.130 *
Santa Marta		0.057 *	0.060 *	0.063 *
Cartagena			0.008 *	0.013 *
Isla Fuerte				0.007 *
** *Nerita tessellata* **				
Cabo de la Vela	0.008 *	0.008 *	0.009 *	0.009 *
Santa Marta		0.005 ^n.s.^	0.006 ^n.s.^	0.003 ^n.s.^
Cartagena			0.005 ^n.s.^	0.004 ^n.s.^
Isla Fuerte				0.005 ^n.s.^

**Table 2 animals-13-02528-t002:** Analysis of molecular variance (AMOVA) evaluating the effects of the two putative barriers on *Acanthemblemaria rivasi*, *Cittarium pica*, and *Nerita tessellata* across the southern Caribbean Sea, Colombia sector: 1. the effect of the Magdalena River plume (MRP); 2. the effect of the absence of a rocky bottom between Cabo de La Vela and Santa Marta (ARB), as well as the almost permanent upwelling of La Guajira (PUG); 3. based on *K* = 3 according to the results of the Bayesian analysis of STRUCTURE. The values of the statistic evaluated the level of differentiation between groups (Φ_CT_), within localities of each group (Φ_SC_), and between all localities (Φ_ST_); the percentage of variation of each source of comparison is indicated; * indicates the significance level *p* < 0.05. ^n.s.^: not significant. PLD: period larval duration.

	*Acanthemblemaria rivasi*		*Cittarium pica*		*Nerita tessellata*	
PLD	<25 Days		<6 Days		>60 Days	
*F* statistic	1. Effect of MRP					
Φ_ST_	0.551 *	44.90%	0.223 *	77.70%	0.129 *	87.10%
Φ_SC_	0.226 *	13.10%	0.159 *	14.70%	0.128 *	12.80%
Φ_CT_	0.420 *	42.00%	0.076 *	7.60%	0.001 ^n.s.^	0.10%
	2. Effect of ARB + PUG					
Φ_ST_	0.582 *	41.80%	0.318 *	68.20%	0.132 *	86.80%
Φ_SC_	0.297 *	17.60%	0.120 *	9.30%	0.127 *	12.70%
Φ_CT_	0.406 *	40.60%	0.224 *	22.40%	0.005 *	0.50%
	3. Based on *K* = 3					
Φ_ST_	0.541 *	45.90%	0.251 *	74.90%		
Φ_SC_	0.090 *	4.60%	0.093 *	7.70%		
Φ_CT_	0.495 *	49.5%	0.174 *	17.40%		

## Data Availability

The data presented in this study are available upon request from the corresponding author.

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
