# Peer review of "A Comparative Phylogeography of Three Marine Species with Different PLD Modes Reveals Two Genetic Breaks across the Southern Caribbean Sea"

_animals, 2023, doi:10.3390/ani13152528_

Round 1
Reviewer 1 Report
Please see the attachment.

-
Author Response
Thanks to the reviewer for providing valuable suggestions on enhancing the manuscript's clarity. We thoroughly considered each suggestion and incorporated them accordingly. In particular, we welcomed the recommendations that aimed to minimize the direct correlation between PLD and gene flow, which we applied to the introduction (L63-73), discussion (L907-914), and conclusions (L1861-1867) sections. Additionally, we acknowledged the importance of understanding the biological and ecological traits of the species in comprehending their phylogeographic structure.
In the introduction, we addressed the taxonomic uncertainty of A. rivasi in Colombia, highlighting that A. aceroi was described as new species by morphological and meristic characteristics, and the molecular analysis was done comparing only Panamanian and Venezuelan COI sequences. Hence, we refrained from recognizing A. aceroi for analysis, as the Colombian samples were not included by Hastings et al. 2020 (L203-209). Nonetheless, we indicated in the discussion that A. aceroi might be present in Colombian waters because we employed a phylogenetic and divergence analysis performed only with COI sequences from Colombian and Panamanian samples deposited in GenBank (see Figure S4 and Table S2). Based on the analyses conducted, it was found that the samples from Santa Marta and Cabo de la Vela differ from those of Panamian, Cartagena, Isla Fuerte, and Capurgana by more than 9% divergence. However, confirming their existence was impossible because there are no Venezuelan COI sequences in GenBank (L916-929). It would have been beneficial to provide more information on this topic, as the ongoing phylogenetic analysis aims to clarify the taxonomy of these two species.
We accepted the suggestion to describe the region's hydrography in the introduction (L172-187) and improved the characterization of the species by providing more biological and habitat-related information (L198-203). We also provided additional information on the statistics implemented in the STRUCTURE analysis (L327-336) and clarified the purpose of determining false positives during the IBD analysis (L411-417). Furthermore, we briefly described the cophenetic.phylo function (L438-442). Finally, we integrated and condensed the 4.2, 4.3, and 4.4 sections in the discussion and conservation implications to improve their clarity (4.3). We also welcomed all suggestions to improve the written ideas' understanding by changing phrases or words. All the accepted changes can be seen in the manuscript and supplementary material.

Reviewer 2 Report
The manuscript (hereafter MS) employs a multiple-species comparative phylogeographic approach to test how biotic and abiotic factors may influence spatial genetic patterns in three species that are characterised by different biological traits. The main result highlights that the interplay between life-history traits and the presence of two putative phylogeographic barriers mainly shapes the spatial genetic structure (SGS) of the three species. Overall, the MS is well written and clearly structured, methods are suitable to reach the study goal and are thoroughly outlined, results are clearly exposed and correctly interpreted, the discussion is sound and not flawed by overstatements or speculations.
The MS certainly deserves to be reccommended for publication pending a bunch of few minor changes authors should take into consideration to further improve it, which are listed below.
Simple summary
L21 - Give full species name here, please.
Introduction
L48 - This is not mandatory, but consider to use gene flow or genetic exchange instead of genetic flow,
L80 - "...because a sampling design needs to be proposed" needs to be slightly rephrased to improve clarity: e.g. "Sampling desing was unsuitable or too coarse to test for this phylogeographic break".
L85 - replace "future" with further or change the verb mode.
L104 - "...that cause spatial patterns of lineages can affect multiple codistributed taxa". Do authors mean lineages within a single species? Please, slightly rephrase to improve clarity.
Methods
L168 - intrallocality -> intra-locality.
L211 - What do authors exactly mean by "understrucured"? Please, explain.
Discussion
L479 - gene flow or genetic exchange.
L493:494 - "which heterogeneous...", I think authors mean "which show heterogeneous...".
L506:507 - "...because the two species evaluated contrast in their larval life histories..." need to be slighlty rephrased to improve clarity; consider, for instance: "the two evaluated species display contrasting larval histories, which could be a signature of the generalized...".
L511 - Replace "exhibits a break" with "exhibit breaks".
L518 - Add genetic or phylogeographic before break.
L520 - Correct the misspelling (ocurrs).
L523 - Please, rephrase "the above demonstrates..." as "The above results demonstrate".
L540 - Why "in the face of a biogeographic barrier"? If authors do not mean "corresponding to a Biogeographic barrier", they should explain what they mean, otherwise they should slightly rephrase this statement.
L541 - Please, erase "could" as the plots either show or not show a linear relationship between geographical and genetic distances.
L544:545 - I think that authors are referring to the spurious effect that barriers may have on IBD testing, whereby significant positive reationship between genetic and geographic distances are the result of barriers ratjer than the decrease of genetic connectivity over increasing distances. This could not be obvious to all readers, therefore, I suggest that stating this more explicitly would improve the clearness and soundness of such concept.
L565 - Please, slightly rephrase "... a K of three populations was..." as "...three populations were...".
L605 - What do authors mean by quietude? tidal regime, strength and variability of marine currents, energy wave, etc.?
L615:617 - This stement would benefit by a slight rephrasing. For instance: "At la Guajira, populations of each species should be considered as genetic management units (GMUs) to prioritize for implementing conservation and fisheries management measures".
L633 - Erase implementation at the end of the sentence.
Figures
Figure 1 - The barplot of Nerita tessellata seems to be cut and population labels are displaced. Additionally, "current principlas" should be replaces by "main currents" in the caption
Figure 4 - In the inset of topleft plot, lineal -> linear
Author Response
Thanks to the reviewer for providing valuable suggestions on enhancing the manuscript's clarity. We thoroughly considered each suggestion and incorporated them accordingly to improve the written ideas' understanding by changing phrases or words. Some changes were not made because the lines indicated they coincided with the improvements suggested by the other reviewer. However, all the accepted changes can be seen in the manuscript and supplementary material.
